# Rapid Visual Detection of Elite Erect Panicle *Dense and Erect Panicle 1* Allele for Marker-Assisted Improvement in Rice (*Oryza sativa* L.) Using the Loop-Mediated Isothermal Amplification Method

**Yonghang Tian** [1,2], **Xiyi Chen** [1,2], **Peizhou Xu** [3], **Yuping Wang** [3], **Xianjun Wu** [3], **Kun Wu** [4], **Xiangdong Fu** [4], **Yaoxian Chin** [1,2,*] **and Yongxiang Liao** [3,*]

1  College of Food Science and Engineering, Hainan Tropical Ocean University, No. 1 Yucai Road, Sanya 572022, China; areckting@hntou.edu.cn (Y.T.); mt48713651@163.com (X.C.)
2  Marine Food Engineering Technology Research Center of Hainan Province, No. 1 Yucai Road, Sanya 572022, China
3  State Key Laboratory of Crop Gene Exploration and Utilization in Southwest China, Rice Research Institute, Sichuan Agricultural University, No. 211 Huiming Road, Chengdu 611130, China; xpzhxj@163.com (P.X.); wyp-sc@163.net (Y.W.); wuxjsau@126.com (X.W.)
4  Institute of Genetics and Developmental Biology, Chinese Academy of Sciences, No. 1 West Beichen Road, Beijing 100101, China; kunwu@genetics.ac.cn (K.W.); xdfu@genetics.ac.cn (X.F.)
*  Correspondence: chinyx1@hntou.edu.cn (Y.C.); 14387@sicau.edu.cn (Y.L.)

**Abstract:** Molecular-assisted breeding is an effective way to improve targeted agronomic traits. *dep1* (*dense and erect panicle 1*) is a pleiotropic gene that regulates yield, quality, disease resistance, and stress tolerance, traits that are of great value in rice (*Oryza sativa* L.) breeding. In this study, a colorimetric LAMP (loop-mediated isothermal amplification) assay was developed for the detection of the *dep1* allele and tested for the screening and selection of the heavy-panicle hybrid rice elite restorer line SHUHUI498, modified with the allele. InDel (Insertion and Deletion) primers (DEP1_F and DEP1_R) and LAMP primers (F3, B3, FIP, and BIP) for genotyping were designed using the Primer3 Plus (version 3.3.0) and PrimerExplore (version 5) software. Our results showed that both InDel and LAMP markers could be used for accurate genotyping. After incubation at a constant temperature of 65 °C for 60 min with hydroxynaphthol blue (HNB) as a color indicator, the color of the LAMP assay containing the *dep1* allele changed to sky blue. The SHUHUI498 rice line that was detected in our LAMP assay displayed phenotypes consistent with the *dep1* allele such as having a more compact plant architecture, straight stems and leaves, and a significant increase in the number of effective panicles and spikelets, demonstrating the effectiveness of our method in screening for the *dep1* allele in rice breeding.

**Keywords:** rice; *dep1*; InDel; LAMP; HNB; SHUHUI498





## 1. Introduction

The breeding of heavy-panicle hybrid rice to further increase rice yield has been gaining research attention [1]. Recently, researchers have incorporated molecular designs in breeding strategies for the development of improved heavy-panicle hybrid rice, leading to the discovery of various beneficial genes for the crop. One such discovery is the pleiotropic gene *DEP1*, which plays an important role in various physiological functions in rice plants, such as yield, nitrogen fertilizer utilization, grain quality, lodging resistance, disease resistance, cadmium tolerance, and drought tolerance [2–4]. The gene is located at chromosome 9 and encodes the G-protein gamma subunit in rice [5]. The mutant *DEP1* allele (*dep1*) is a gain-of-function variant with a 625 bp deletion and a 12 bp insertion in exon 5, which causes the formation of dense and erect panicles by allowing for an increased

number of primary and secondary branches and, thus, grain number per panicle in rice. The allele is predominantly found in the *japonica* subspecies [5,6] and was introduced into the elite *indica* restorer line 9311, significantly increasing the yield and grain number per panicle [7]. The allele also enables denser planting and higher yields by improving canopy structure and increasing plant photosynthetic efficiency [8,9]. Research has shown that the *dep1* allele improves the harvest index and yield by improving the efficiency of nitrogen application [10], thereby helping to reduce the carbon footprint and promote a sustainable rice industry. In addition, dep1 interacts with LPA1 to activate the expression of *PIN1a*, which improves rice resistance to blights [4], minimizing the need for pesticides and, thus, ensuring stable yields. At the same time, the allele also improves plant tolerance to Cd (Cadmium) [11] and allows plants to survive harsh environments by improving their tolerance to drought, cold, and salinity [12,13]. Furthermore, *dep1* is able to modulate the balance of the carbon and nitrogen metabolism of rice, contributing to improved grain quality [3,14]. Therefore, the *dep1* allele has great potential in the development of green super rice [15,16] and warrants further research. As such, the development of convenient on-site detection for *dep1* would considerably enhance our ability to monitor the effects and success of *dep1* applications in rice breeding.

Several reports have been presented on the detection of the *dep1* allele using PCR (polymerase chain reaction) technology based on molecular markers such as InDel-E5 [17], *DEP1*S9 [18], *DEP1*E5ID [19], and H90 [20]. However, these methods require high temperatures and specialized devices for data visualization, rendering them impractical for on-site testing [21,22]. LAMP (loop-mediated isothermal amplification) is an isothermal nucleic acid detection technique [23,24] that is able to achieve an accuracy similar to PCR. Moreover, this technology does not require specialized or sophisticated instruments while also offering the rapid amplification of targeted DNA with high efficiency, specificity, and sensitivity. LAMP technology is also user-friendly, as it can be easily mastered and performed by laymen without the need for prior molecular experiences [25]. Recently, LAMP has been successfully used for the rapid detection of transgenes [26], blight-resistant genes [27], and authenticity [28] in rice plants.

In this study, we aim to establish a colorimetric LAMP assay for the visual detection of the *dep1* allele in the heavy-panicle elite hybrid rice restorer line SHUHUI498. The outcome of this study could provide a technical reference for the development and application of visual markers for other important functional genes. The LAMP molecular marker's design takes advantage of *dep1's* 625 bp deletion and a 12 bp insertion in exon 5, which is a conserved region of the *DEP1* gene and, therefore, enables specific differentiation between the two alleles.

## 2. Materials and Methods

### 2.1. Plant Material and Genomic DNA Isolation

The rice varieties used in the experiment consisted of 10 rice lines, namely, WYJ7, WYJ3, SN265, QCL2, N580, N130, Nip, SHUHUI498, ZH11, 9311, and the progenies of WYJ7 backcrossed with SHUHUI498, all provided by the Rice Research Institute of Sichuan Agricultural University.

Genomic DNA from rice leaves at the heading stage was extracted using the CTAB method [29]. The extracted DNA templates were stored at −20 °C for subsequent use.

### 2.2. The InDel and LAMP Marker Design

The DNA sequence for the rice *dep1* allele was downloaded from the RGAP (Rice Genome Annotation Project) [30].

After the alignment and analysis of the *dep1* allele sequence against the NCBI nucleotide database (https://blast.ncbi.nlm.nih.gov/Blast.cgi, accessed on 15 May 2019), the deletion and insertion variants [5] that include both upstream and downstream conserved sequences in exon 5 of *dep1* were taken as target sequences for the design of InDel primers using Primer3 [31]. The sequence was uploaded to PrimerExplorer V5

(http://primerexplorer.jp/lampv5e/index.html, accessed on 1 January 2024), and LAMP primers were designed with one of the primers spanning the 625 bp→12 bp mutated locus in exon 5 of the *dep1* variant to ensure specificity. After alignment was carried out in EnsemblPlants (https://plants.ensembl.org/Multi/Tools/Blast, accessed on 1 January 2024), generated primers with the highest specificity were selected. The primers were then synthesized by Sangon Biotech (Shanghai) Co., Ltd., Shanghai, China.

### 2.3. PCR Amplification and Detection

The PCR reaction was carried out in a reaction volume of 20 μL containing 2 μL of forward and reverse primer (10 μM), 4 μL of a 2 × Taq PCR mix, 10 μL of ddH$_2$O, and 2 μL of DNA templates. PCR amplification was performed on a Bio-Rad T100™ Thermal Cycler (Bio, Hercules, CA, USA) with the following conditions: initial denaturation at 94 °C (3 min); 30 cycles of denaturation (94 °C, 30 s), annealing (56–63 °C, 30 s), and extension (72 °C, 1 min); and final extension (72 °C, 5 min). The presence of the amplification products was verified via agarose gel electrophoresis (BIO-OI gel imaging system, Guangzhou Guangyi Biotechnology Co., Ltd., Guangzhou, China). DNA templates containing the *dep1* allele will generate smaller amplicons (409 bp), whereas templates containing the *DEP1* allele will produce larger amplicons (1034 bp).

### 2.4. LAMP Amplification and Colorimetric LAMP Assay

#### 2.4.1. LAMP Amplification and Agarose Gel Electrophoresis Detection

The LAMP reaction system was prepared as described in the kit's manual (2× Lamp PCR Master Mix, B532455, Sangon Biotech (Shanghai) Co., Ltd., Shanghai, China) and adjusted for a 12.5 μL reaction volume: 6.25 μL of 2 × LAMP Mix Buffer, 10 μM F3/B3 primers (each 1 μL), 10 μM FIP/BIP primers (each 0.25 μL), 0.5 μL DNA template, 0.25 μL DNA polymerase (0.16 U/μL).

The LAMP reaction was carried out according to kit instructions with some modifications. Briefly, the target gene was denatured at 95 °C for 5 min, followed by incubation at a constant temperature (55–65 °C) for 1 h. The enzyme activity was then inactivated at 80 °C for 10 min, and the LAMP reaction was terminated by holding it at 12 °C for 5 min. The presence of LAMP products was verified with agarose gel electrophoresis, as described in Section 2.3.

#### 2.4.2. Colorimetric LAMP Assay

The reaction system was prepared according to the manufacturer's instructions (LAMP HNB Amplification Kit, A3802, HaiGene Biotech Co., Ltd., Harbin, China), adjusted for a 12.5 μL reaction volume. The LAMP amplification protocol was performed as described in Section 2.4.1. After the completion of the amplification reaction, the color of the reaction solution was observed with the naked eye for changes [32]. The reaction solutions that contained the *dep1* allele changed to a sky-blue color, while the reaction solutions without it remained violet.

### 2.5. Application of LAMP Colorimetric Assay in the Improvement of the Heavy-Panicle Elite Hybrid Rice Restorer Line SHUHUI498

The potential application of our designed LAMP assay was tested via screening for heavy-panicle elite hybrid rice restorer line SHUHUI498 introduced with the *dep1* allele. Cross-breeding was first performed by using *dep1*-carrying WYJ7 [33] rice as the maternal parent and SHUHUI498 [34], a heavy-panicle elite hybrid rice restorer line, as the paternal parent. The resultant progenies of the cross were backcrossed twice (BC$_1$ and BC$_2$) with SHUHUI498 as the recurrent parent, followed by consecutive self-pollinations (F$_n$), during which elite progenies in each generation (n = 50) were screened and selected to establish a stable superior line. The screening and selection of superior individuals in generations of BC$_1$F$_1$ and BC$_2$F$_1$ were performed by observing phenotype characteristics supplemented by genotyping using the colorimetric LAMP assay described above. Individual plants

carrying *dep1*, which is a dominant negative gene [5], are typically distinguished by plant architecture, panicle, leaf, and grain shape. As 87.5% of the WYJ7 genetic background would theoretically be eliminated in $BC_2F_1$ via self-pollination, an experienced breeder could achieve more than 95% clearance of the genetic background when carrying out phenotype selection in tandem with genotyping [35]. After progenies carrying the *dep1* allele in $BC_2F_1$ were verified with the LAMP assay, the successive selection of excellent individuals from each segregating generation was carried out until a stable line (tentatively named SHUHUI498$^{dep1}$) was established, and both its genotypes and phenotypes were analyzed.

Forceps emasculation (FE) was used to produce crosses and backcrosses [36]. Superior progenies in the backcross populations were screened and selected for hybridization using on-field performance, phenotype evaluation, and genotyping of *dep1* allele. Elite individuals that flowered during the heading stage were collected before 8 a.m. each day. Using flowering time as an indicator, both bloomed and immature florets were cut off and discarded, while the remaining florets were used for forceps emasculation on the next day [37]. After cutting off about 1/3 of the upper part of the spikelet diagonally with a pair of shears, the exposed stamens were wholly removed with tweezers before covering the panicles with hybridization bags. Panicles of SHUHUI498 that were emitting or on the verge of emitting pollen were collected around 10 a.m. to pollinate the emasculated spikelets, which were then bagged and recorded. Hybridized grains were collected at the end of 20 days with a grain count of approximately 100 grains, with sufficient grain counts obtained by re-hybridizing if necessary.

Twenty individuals of SHUHUI498 were grown per generation as controls, while 50 progenies were used for screening or backcrossing. Planting was carried out in ten plants per row at 30 cm × 15 cm spacing. WYJ7, SHUHUI498, and their progenies were planted in Lingshui District in Hainan Province during the winter and in Wenjiang District in Sichuan Province during the summer.

Agronomic traits such as PH (plant height), EPN (effective panicle number), PL (panicle length), GP (grains per panicle), SP (spikelets per panicle), and YP (yield per plant) were investigated in the stabilized lines according to the literature [5,10]. Genotyping was carried out according to the steps described in Section 2.4.

## 3. Results

### 3.1. InDel and LAMP Markers

The InDel primers (DEP1-F and DEP1-R) for genotyping the *dep1* allele were designed using the Primer3Plus (version 3.3.0, Figure 1, Table 1). The sizes of the PCR amplification products of the *dep1* and *DEP1* alleles using the InDel marker were 409 bp and 1034 bp, respectively. We also included an InDel marker that is reported elsewhere (Table 1) for result comparison and verification [17].

The LAMP primers F3, B3, FIP, and BIP were designed using PrimerExplorer V5 for genotyping the *dep1* allele (Table 1), specifically targeting six highly conserved regions of the *DEP1* gene (Figure 1). Except for differences in the 12 bp insertion and 625 bp deletion between the *DEP1* and *dep1* alleles at exon 5, the six gene segments were consistent across the approximately 600 published genomes (Supplementary Data S1.out and S2.out), revealing them as highly conserved regions. The distance between the 5′ end of B2 and the 5′ end of B1 (or the 5′ end of F2 and the 5′ end of F1) determined the size of the loop formed in the reaction, which, in turn, affected the LAMP amplification efficiency [38]. In this study, the most suitable distance was found to be 40~60 bp. Hence, the distance between B1 and B2 was set at 57 bp, which enabled the LAMP primers to accurately bind to the DNA template and amplify efficiently. The 625 bp→12 bp mutation between B2 and B1 in the *DEP1* allele severely impaired the loop formation in the reaction, thus preventing the LAMP reaction.

**Table 1.** Primers' sequences of InDel and LMP markers.

| Primers | Sequences (5′-3′) | Usage | Product Size |
|---|---|---|---|
| DEP1-F | TAAGCCAAACTGCAGTGCG | Forward Primer for InDel Marker | *dep1* (409 bp), *DEP1* (1034 bp) |
| DEP1-R | GTTCAACCTCGTCTCATAGCT | Reverse Primer for InDel Marker | |
| F3 | CATGCTGTAGTCCAGACTG | Forward Outer Primer for LAMP Marker | |
| B3 | AAGCAACCACTGAGACAG | Backward Outer Primer for LAMP Marker | |
| FIP (F1C-F2) | TTCGGTTTGCAGCAAGAAGG-CTGCTCATGCTGTAAACCTA | Forward Inner Primer for LAMP Marker | |
| BIP (B1C-B2) | TGCGATACATCGTGCTGCAA-GGGCATCGACAACCCA | Backward Inner Primer for LAMP Marker | |
| InDel-E5-F [17] | TCCAGGGATGTAATCATCTTTGTT | Forward Primer for InDel Marker InDel-E5 | *dep1* (733 bp), *DEP1* (1357 bp) |
| InDel-E5-R [17] | GGCTCCATATCTTCACGGTCTA | Forward Primer for InDel Marker InDel-E5 | |

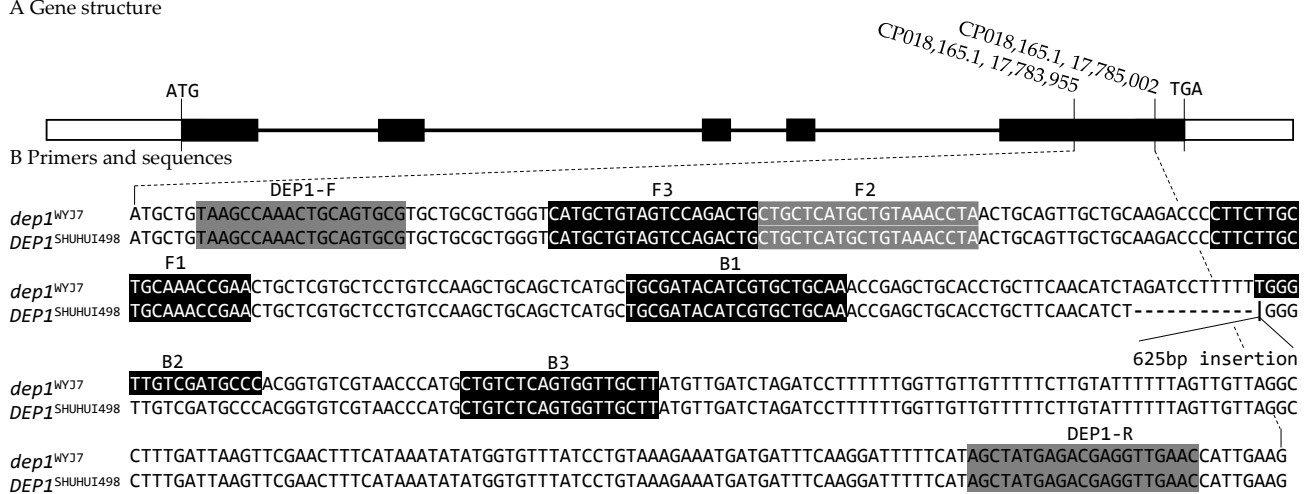

**Figure 1.** Schematic of the *DEP1* gene structure and primer locations. (**A**) Gene structure. ATG denotes the start codon, TGA denotes the stop codon position, the white box before ATG denotes the 5′UTR, the white box after TGA denotes the 3′UTR, and the black box in the middle denotes the exon. (**B**) Primers and sequences. *DEP1*^WYJ7 indicates that the *dep1* gene sequence is from the rice variety WYJ7, while *DEP1*^SHUHUI498 indicates a *DEP1* gene from the SHUHUI498 variety. "CP018,165.1, 17,783,955" is indicated as base "A" at positions 17, 783, and 955 bp of GenBank accession number CP018,165.1, "CP018,165.1, 17,785,002" is the same. The sequences shaded in dark gray under the InDel marker primers DEP1-F and DEP1-R denote the primer sequences and their positions on the genes. The shaded characters under F3, F2, F1, B1, B2, and B3 denote the six regions of the gene targeted by the LAMP primers.

### 3.2. Optimizing PCR and Detecting InDel Markers in Different Rice Varieties

The PCR amplification annealing temperatures of primers DEP1-F and DEP1-R were first optimized to ensure accuracy (Figure 2). The results showed that unique bands of approximately 1000 bp in size could be amplified via PCR using SHUHUI498's DNA as a template at different annealing temperatures ranging from 56.0 °C to 63.0 °C. The targeted bands produced in each lane were clear, indicating excellent primer specificity. An annealing temperature of 63.0 °C, which provided better amplification, was finally chosen as the optimized annealing temperature for PCR amplification.

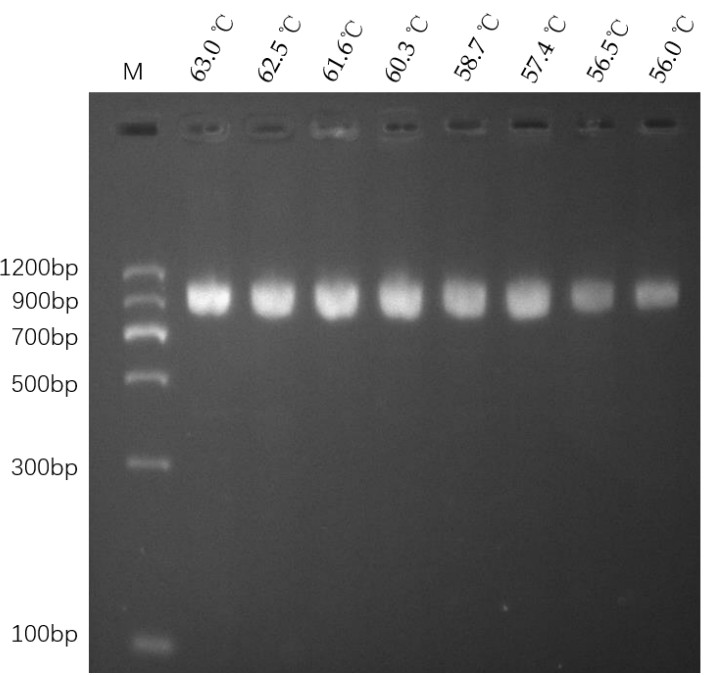

**Figure 2.** PCR amplification at different annealing temperatures. Lane M is a DNA ladder (MD102, Tiangen Biotech (Beijing) Co., Ltd., Beijing, China). Lanes 1 to 8 correspond, respectively, to annealing temperatures of 63.0 °C, 62.5 °C, 61.6 °C, 60.3 °C, 58.7 °C, 57.4 °C, 56.5 °C, and 56.0 °C, in that order.

Following the optimized PCR conditions, DNA materials from rice lines WYJ7, WYJ3, SN265, QCL2, N580, N130, Nip, SHUHUI498, ZH11, and 9311 were used as templates for PCR amplification, and the resultant amplicons were verified via electrophoresis on a 2% agarose gel (Figure 3). The results showed that the WYJ7, WYJ3, SN265, QCL2, N580, and N130 rice lines, all of which carry the *dep1* allele, produced smaller amplicons, while the *DEP1*-carrying rice lines Nip, SHUHUI498, ZH11, and 9311 produced larger amplicons. Therefore, the sizes of the PCR amplicons are in accordance with the genotypes.

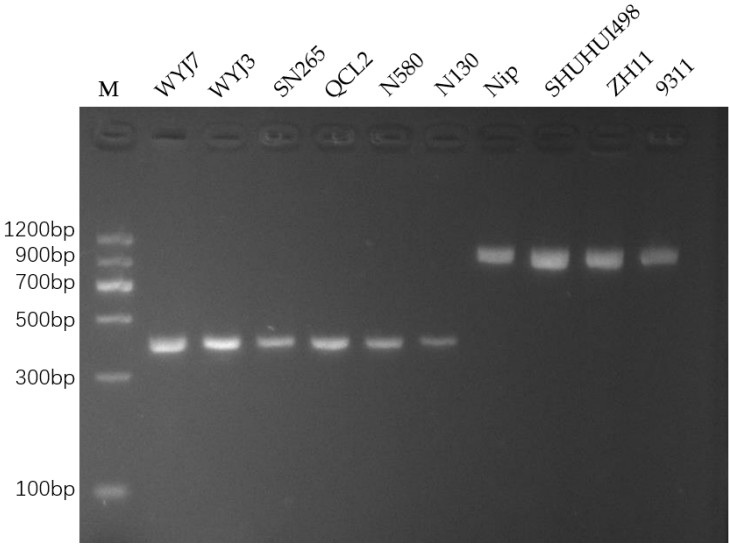

**Figure 3.** Electrophoretic detection of PCR-amplified products from different rice varieties. Lane M is a DNA ladder (MD102, Tiangen Biotech (Beijing) Co., Ltd., Beijing, China), and the varieties in lanes 1 to 10 are WYJ7, WYJ3, SN265, QCL2, N580, N130, Nip, SHUHUI498, ZH11, and 9311, in that order.

### 3.3. Optimization of LAMP Reaction Temperature

After the LAMP reaction was carried out at different temperatures ranging from 55 °C to 65 °C using the DNA template from WYJ7, the resultant products were verified using 2% agarose gel electrophoresis (Figure 4). The results showed typical ladder-like bands exhibited by LAMP amplification products. When the temperature reached 56.9 °C, the bands were blurred and dull, indicating few LAMP products. At 65.0 °C, 64.3 °C, 63.0 °C, 61.1 °C, and 58.8 °C, the bands were clearer and brighter, indicating more amplification products. Hence, 65.0 °C was chosen as the optimized LAMP reaction temperature for the following experiments.

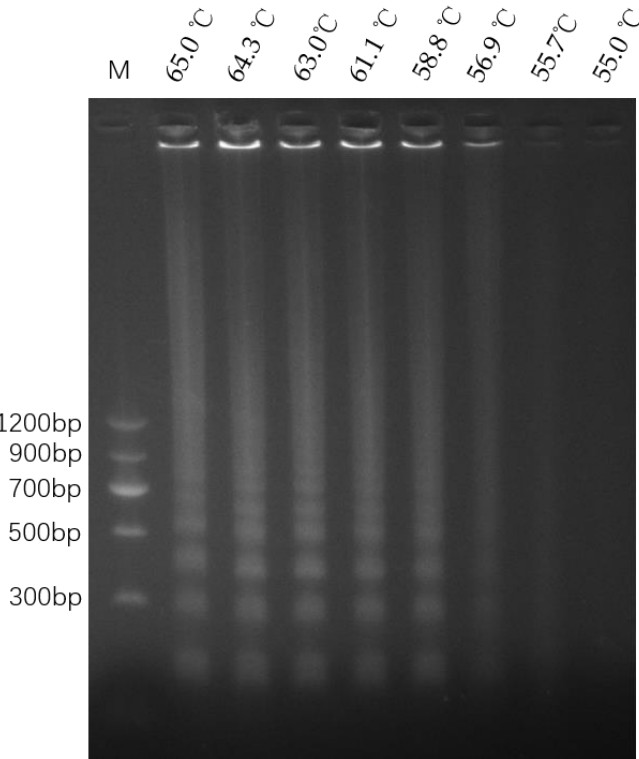

**Figure 4.** Electrophoresis of LAMP products at different temperatures. Lane M is a DNA ladder (MD104, Tiangen Biotech (Beijing) Co., Ltd., Beijing, China). Lanes 1 to 8 correspond to the respective LAMP reaction temperatures of 65.0 °C, 64.3 °C, 63.0 °C, 61.1 °C, 58.8 °C, 56.9 °C, 55.7 °C, and 55.0 °C, in that order.

### 3.4. LAMP Reaction and Colorimetric LAMP Assay

LAMP amplification and electrophoresis were performed under LAMP-optimized conditions using DNA from the aforementioned 10 different rice lines as templates (Figure 5A). The gel electrophoresis results showed characteristic ladder bands in the lanes of WYJ7, WYJ3, SN265, QCL2, N580, and N130, which are rice lines that carry the *dep1* allele. In contrast, no LAMP amplification products were found in the lanes representing the *DEP1*-containing varieties (Nip, SHUHUI498, ZH11, and 9311). These results were expected, as the LAMP markers were designed using the DNA sequence of the *dep1* allele, meaning only the *dep1*-containing DNA template would be amplified. Nip, SHUHUI498, ZH11, and 9311 are of the *DEP1* genotype, which has a large insertion on the DNA template corresponding to the primer's BIP-matching region, which impairs the formation of the loop in the LAMP reaction, thus resulting in no amplicons being produced.

A colorimetric LAMP assay was also performed with the aforementioned 10 rice varieties (Figure 5B). The results showed that the tubes containing DNA templates of WYJ7, WYJ3, SN265, QCL2, N580, and N130 successfully completed the LAMP reaction, turning the solution sky blue. In contrast, the tubes containing the DNA templates of

Nip, SHUHUI498, ZH11, and 9311 did not undergo a LAMP reaction, and the solution remained violet.

At present, the genomic information of the eight rice varieties used in this study, except for N580 and N130, are published in open databases. The genotypes of WYJ7, WYJ3, SN265, QCL2, N580, and N130 were all confirmed to be of *dep1*, while the genotypes of Nip, SHUHUI498, ZH11, and 9311 are *DEP1*. Hence, the results of the InDel markers, LAMP amplification, and LAMP colorimetric assay in this study were all consistent with the published genotypic data.

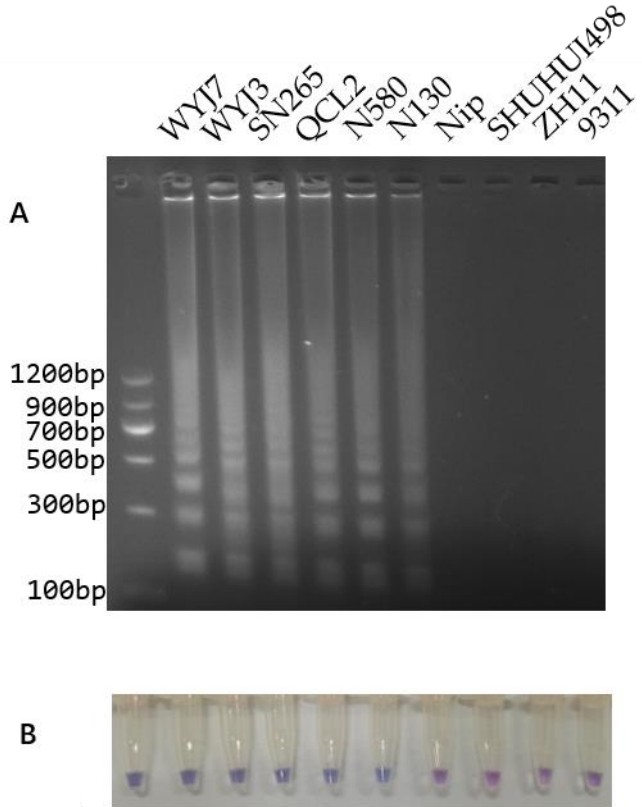

**Figure 5.** LAMP amplification and visual colorimetric assay for the detection of different rice varieties. (**A**) Electrophoretic detection of LAMP amplification products. (**B**) Visual colorimetric LAMP assay. Lane M is a DNA ladder (MD114, Tiangen Biotech (Beijing) Co., Ltd., Beijing, China). The DNA templates used for each of the lanes are the same as those in Figure 3.

### 3.5. Application of the Colorimetric LAMP Assay in Breeding an Improved Heavy-Panicle Elite Hybrid Rice Restorer Line

The SHUHUI498$^{dep1}$ rice line was successfully bred by introducing the *dep1* allele into the heavy-panicle elite hybrid rice restorer line SHUHUI498 (Figures 6 and 7). SHUHUI498$^{dep1}$ was compact, with upright leaves and stems, and had a reduced plant height of about 14 cm (Figures 6B and 7A). The plants grew uniformly in the field (Figure 6D) with shorter panicles and increased grain density (Figures 6B,C and 7C). Effective panicles per plant (Figures 6B and 7B) and spikelet numbers per panicle were significantly increased in SHUHUI498$^{dep1}$ (Figure 7E), although the grain number per panicle and the yield per plant did not (Figure 7D,E). This is probably due to the low light and low day/night temperature difference in southwest China, which is unfavorable for organic matter accumulation, leading to insufficient photosynthetic products being produced to sustain the increase in the actual grain numbers per panicle and yield, despite the increase in effective panicles per plant and spikelet numbers per panicle in SHUHUI498$^{dep1}$. This may also be the reason why *dep1* is mainly found in *japonica* rice [5]. On the other hand, SHUHUI498$^{dep1}$ has the potential to achieve high yields in high-light environments,

which can boost its photosynthetic efficiency and allow for the increased planting density necessary to raise the yield per unit area [8,9].

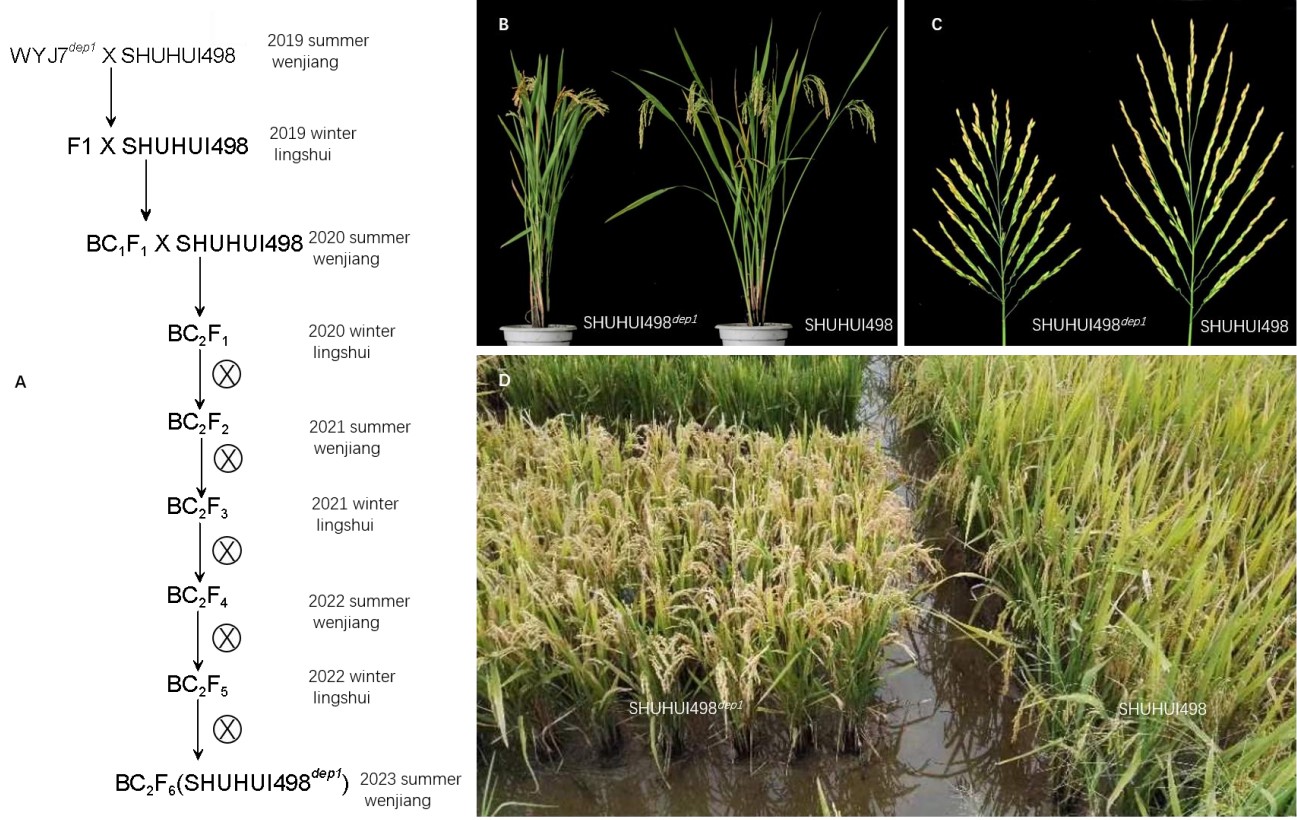

**Figure 6.** Breeding plan and phenotypes of SHUHUI498*dep1*. (**A**) Breeding plan of SHUHUI498*dep1*. (**B**) Morphology comparison between SHUHUI498*dep1* and SHUHUI498. (**C**) Comparison of panicle traits between SHUHUI498*dep1* and SHUHUI498. (**D**) Field performance of SHUHUI498*dep1* and SHUHUI498. Symbols: ×, cross; ⊗, self-cross.

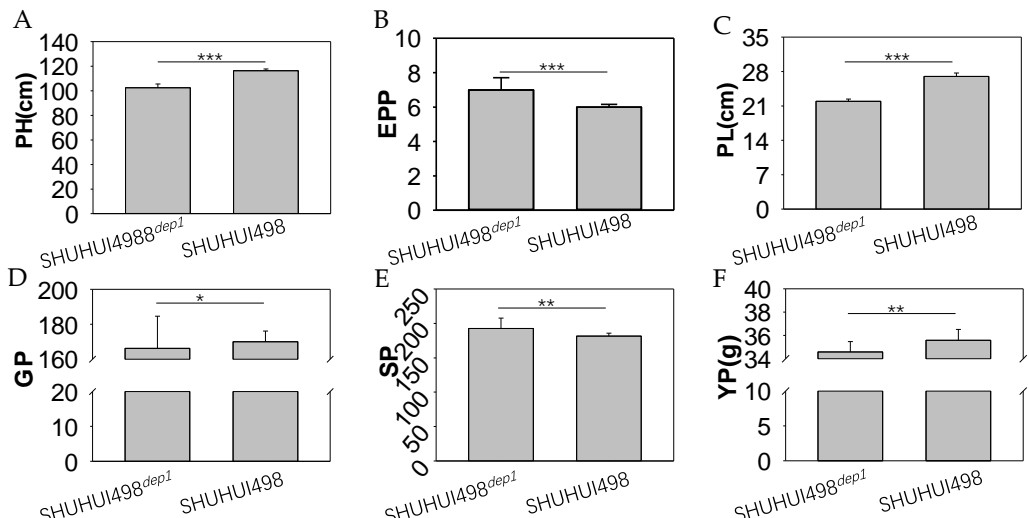

**Figure 7.** Analysis of key agronomic traits between SHUHUI498*dep1* and SHUHUI498. (**A**) PH, plant height. (**B**) EPP, effective panicles per plant. (**C**) PL, panicle length. (**D**) GP, grain number per panicle. (**E**) SP, spikelet number per panicle. (**F**) YP, yield per plant. Analysis of the data was performed using a two-tailed Student's *t*-test. Significances are denoted as *** ($p < 0.001$), ** ($p < 0.01$), and * ($p < 0.05$).

SHUHUI498$^{dep1}$, SHUHUI498, and the progenies of SHUHUI498 were genotyped with the InDel markers (InDel-E5) reported elsewhere [17] (Figure 8A). The results show that the genotype of SHUHUI498 is *DEP1*, while SHUHUI498$^{dep1}$; individuals 3, 5, and 6 in BC$_1$F$_1$; and individuals 8 and 9 in BC$_2$F$_1$ carry the *dep1* allele. This is consistent with the results in both the gel electrophoresis and colorimetric assay of LAMP amplification (Figure 8A). Therefore, the colorimetric LAMP assay that we developed is able to accurately detect the targeted genotype in SHUHUI498$^{dep1}$, SHUHUI498, and their progenies (Figure 8B,C), which, when combined with the phenotypic characteristics of the *dep1* gene, can be used for the rapid on-site screening of target individuals in rice breeding programs (Figure 7).

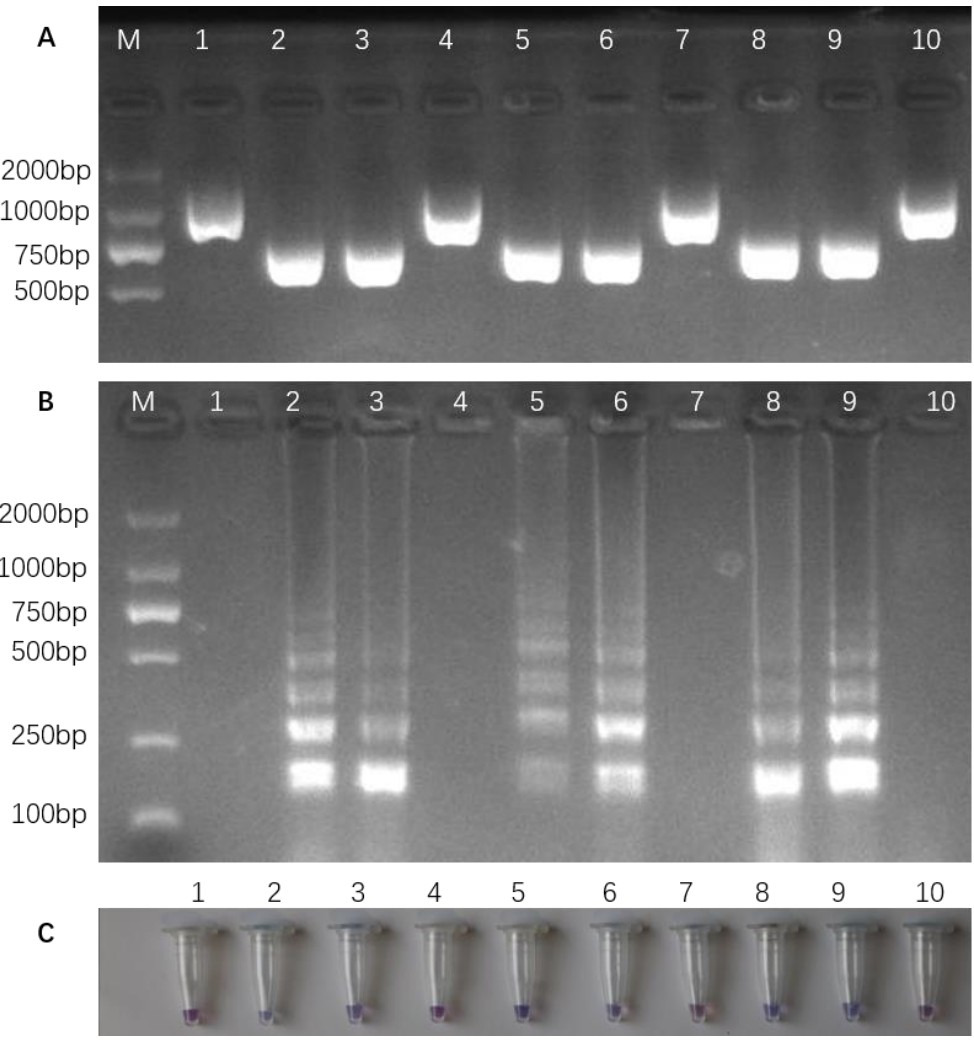

**Figure 8.** Genotyping of SHUHUI498$^{dep1}$, SHUHUI498, and their offspring. (**A**) Electrophoresis of the PCR amplification products with the InDel marker InDel-E5 [17]. (**B**) Electrophoresis of LAMP amplification products. (**C**) Visual colorimetric LAMP amplification. Lane M is a DNA ladder (MD114, Tiangen Biotech (Beijing) Co., Ltd., Beijing, China). The DNA templates in lanes 1 to 10 and tubes 1 to 10 are as follows: 1 to 2 are SHUHUI498 and SHUHUI498$^{dep1}$, 3 to 6 are individuals in BC$_1$F$_1$, and 7 to 10 are individuals in BC$_2$F$_1$.

## 4. Discussion

Molecular marker-assisted breeding (MAB) is a valuable technique for producing rice varieties with improved yield, quality, and disease resistance [39,40]. The technique has the advantages of being convenient, highly efficient, and unaffected by environmental conditions, making it possible to rapidly improve desired traits and develop superior varieties. The application of MAB has overcome many of the limitations faced in traditional

breeding, enabling the development of high-quality improved lines in a relatively short period. Screening and selection in conventional breeding programs are usually achieved empirically by observing phenotypes, from the screened parental plants to the hybridized progenies, and success is more often than not due to the experience of the breeders. Hence, traditional breeding programs tend to require a long breeding cycle and, subsequently, have low efficiency. Following the booming development in functional genomics, many functional genes have been discovered [41]. The incorporation of these functional genes into commercial breeding is a crucial step in crop research [42]. With the aid of molecular markers, the screening and selection of desired traits in a breeding program can be greatly enhanced through the definite confirmation of genotypes even before phenotype observation is possible. In addition, MAB enables the simultaneous selection of targeted genes and their neighboring genes (both upstream and downstream), which improves the breeding accuracy, speeds up the removal of genetic background, and reduces linkage tedium, leading to a shortening of the breeding cycle. Thus, MAB is currently the most popular tool used by most breeders. Although molecular markers such as InDel, CAPs (Cleaved Amplified Polymorphic Sequences), and dCAPs (Derived Cleaved Amplified Polymorphic Sequences) are commonly used in both academic research and commercial breeding given their technical stability and precision, these methods are, nevertheless, complex and laborious, including multiple steps such as DNA extraction, PCR amplification, restriction enzyme digestion, gel electrophoresis, and imaging electrophoretic results. Furthermore, conventional PCR methods—such as RFLP (Restriction Fragment Length Polymorphism), AFLP (Amplified Fragment Length Polymorphism), SCAR (Sequence-Characterized Amplified Regions), SSR (Simple Sequence Repeats), RAPD (Random Amplified Polymorphic DNA), ISSR (Inter Simple Sequence Repeats), TRAP (Telomeric Repeat Amplification Protocol), ARMS (Amplification Refractory Mutation System PCR), KASP (Kompetitive Allele Specific PCR), and HRM (High-Resolution Melting curve)—require specialized equipment for thermal cycling and gel electrophoresis [43–45], which incurs extra cost. As such, isothermal methods such as LAMP represent attractive options for practical applications given their ease of use; sensitivity; reasonable accuracy; convenience; and, once developed, low cost [46]. Even though LAMP protocols are prone to false positives because of their high sensitivity, an established LAMP protocol, such as the one shown in this study, when combined with the rapid extraction of DNA templates [26,47] with commercially available kits, can enable a breeder to determine the targeted genotypes on-site with a simple and straightforward protocol at a reasonable cost. This is particularly appealing to rice breeders in developing countries, many of whom are farmers who are poorly educated with little to no knowledge of molecular biology and who are more concerned about cost than accuracy. This demand can be reflected by the fact that more than 60% of the molecular diagnostics kits available on the market are LAMP-based and are widely used in the identification of viruses, bacteria, and fungi, as well as medical diagnosis and food safety. The results of our study indicate that our developed LAMP assay has comparable precision and sensitivity to standard PCR using InDel markers (Figures 3 and 8A), further lending credence to the potential of colorimetric LAMP assays.

The need for global food security and environmental concerns have driven the research community to find the right balance between high yield and the use of nitrogen fertilizers [48]. In order to reduce the conflict between food resources and the environment, scientists have proposed the concept of green super rice, which focuses on the discovery and application of nitrogen-efficient genes [49]. Research has found that the *dep1* allele can increase the harvest index and yield by improving nitrogen use efficiency [10] and regulating rice quality, disease resistance, stress resistance, and so on. Hence, *dep1* is considered a key allele for breeding green super rice varieties [7,50]. The colorimetric LAMP assay for *dep1* allele designed in this study, combined with rapid DNA extraction methods [26,47], presents a convenient method for the on-site genotyping of *dep1*, contributing greatly to the breeding and development of green, high-yielding rice varieties.

Plant architecture is one of the key factors influencing yield and has received much attention in rice breeding. Numerous alleles, such as *dep1*, *sd1*, *IPA1*, *d14*, and *d61*, have been found to regulate plant architecture and, thus, have been widely cloned and used in rice breeding [51]. The *dep1* allele is widespread in *japonica* rice varieties, with more than half of the cultivated rice varieties in northeast China carrying this allele [6]. Rice lines WYJ7, WYG3, and LG31, all of which carry the *dep1i* allele, are among the most widely cultivated rice varieties. In addition, although *indica* rice varieties carry the *DEP1* allele natively, it has been reported that introducing the *dep1* allele into *indica* rice can significantly raise its yield [7]. The data from this study also showed that both the effective panicles per plant and the spikelet numbers per panicle were significantly increased after the successful introduction of the *dep1* allele into the heavy-panicle elite hybrid rice restorer line SHUHUI498 (Figures 6 and 7). Therefore, the colorimetric LAMP assay for the *dep1* allele developed in this study is a valuable screening tool for targeted gene improvement in MAB for rice.

Since global warming increases the threat of pests and diseases, breeding new varieties with improved stress and disease resistance is important to securing stable rice yields as global temperature rises [52]. The incorporation of newly discovered disease and stress resistance genes into rice production has shown satisfactory results and represents a viable strategy [53]. The colorimetric LAMP assay for *dep1* detection developed in this study can also be adapted and modified as a convenient way to screen and breed new disease- and stress-resistant rice varieties to ensure stable yields and food security.

It is important to note that, while the results of our study are in agreement with each other and are fully aligned with our expected outcome, the experiments were nevertheless conducted in a controlled environment with known genotypes. For the assay to be applicable for field use, which involves many unknown variables, internal and standard controls have to be developed to ensure accuracy, including the prevention of false-negative results.

## 5. Conclusions

In this study, we established a convenient colorimetric LAMP assay for genotyping the *dep1* allele in rice based on *dep1*-specific sequences available in the published literature and databases. The developed method proved to be accurate and specific, suitable for the on-site screening of the *dep1* allele. We further confirmed the feasibility of introducing the *dep1* allele into the heavy-panicle elite hybrid rice restorer line SHUHUI498 and showed its potential to increase yield. This study not only provides technical support for genotyping *dep1*, an allele for dense erect panicles and high yields in rice but also serves as a reference for the future development and application of LAMP-functional markers for other important genes. For further study, we will investigate the incorporation of additional loop primers into our LAMP assay in order to boost its efficiency.

**Supplementary Materials:** The following supporting information can be downloaded at: https://www.mdpi.com/article/10.3390/cimb46010032/s1, Data S1: Schematic comparison of the *DEP1* from Nipponbare_IRGSP_4.0_AP008215.2 with the *dep1* from 13 other varieties; Data S2: *DEP1* comparisons across nearly 600 different genomes.

**Author Contributions:** Y.T. and Y.L. conducted the bioinformatic analysis and primer design. Y.W., P.X. and X.W. confirmed the primers with experiments and breeding. X.C. conceptualized and wrote the first draft. K.W., Y.C. and X.F. designed the experiments and revised the manuscript. All authors discussed the results and contributed to the manuscript. All authors have read and agreed to the published version of the manuscript.

**Funding:** This study was supported by the National Natural Science Foundation of China, Grant No. 31960399; the Natural Science Foundation of Hainan Province, Grant No. 321MS050; and funding from the Chengdu Science and Technology Bureau, Grant No. 2022-YF09-00036-SN.

**Institutional Review Board Statement:** Not applicable.

**Informed Consent Statement:** Not applicable.

**Data Availability Statement:** Additional data are available from the corresponding authors upon request.

**Conflicts of Interest:** The authors declare no conflicts of interest.

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
