# Peer review of "Rapid Visual Detection of Elite Erect Panicle Dense and Erect Panicle 1 Allele for Marker-Assisted Improvement in Rice (Oryza sativa L.) Using the Loop-Mediated Isothermal Amplification Method"

_cimb, doi:10.3390/cimb46010032_

Round 1

Reviewer 1 Report

Comments and Suggestions for Authors

The manuscript (cimb-2776571) entitled “Rapid Visual Detection of Elite Erect Panicle Allele dep1 for Marker-Assisted Improvement in Rice (Oryza sativa L.) Using  Loop-Mediated Isothermal Amplification Method” This study established a convenient colorimetric LAMP method for genotyping  dep1 gene in rice based on dep1-specific sequences available in published literature and databases. The developed method was proven accurate and specific, suitable for on-site screening of dep1 gene and helps in MAB of rice. The manuscript is well drafted, except few grammatical errors  and may be considered after a satisfactory reply and compliance with the observations.

Figures

Line 73- Remove et al.

Line 169 and Line 205- Both the images Figure 1 and Figure 2 are missing in the text, incorporate them

Line 205- Different annealing

Line 302-Figure 7 & Line 322-Figure 8: Figure title needs to be replaced and revise the title

351-354- In addition to InDel, CAPs and dCAPs, other molecular methods include RFLP, AFLP, 351 SCAR, SSR, RAPD, ISSR, TRAP, ARMS: To highlight the LAMP assay you cannot blindly conclude that SSR and other markers are more laborious etc. still SSR markers are used widely throughout for MAB in rice. Rewrite the sentence 351-354

Line 352 -353-KASP and HRM, many of which are laborious, complex, difficult to design and develop, unstable, requires specialized equipment, expensive and inconvenient for in-field detection. No wrong information, the most used assay for MAB is KASP and it is much easier than LAMP. Rewrite with references, since you have not used any other methods except LAMP, you cannot compare and justify it.

The discussion part needs to be improved

Has this method been practically demonstrated to the rice breeders, if yes, highlight it in the text.

All the figures need to be checked properly and respective titles should be revised

Comments on the Quality of English Language

Good

Author Response

Authors' Responses to Reviewer's Comments

Reviewer 2 Report

Comments and Suggestions for Authors

The reviewed manuscript is dedicated to design and validation of a LAMP-based protocol for the depi1 allele of rice. The presented results are interesting for scientists, specializing on the field of molecular diagnostics and would help in selection of novel superior rice varieties. However, a number of issues needs to be addressed before publication.

Major issues:

1.      Thoughtful language editing is necessary to improve quality of the manuscript. Currently, some sentences are confusing. Also, terms in the manuscript need to be corrected for better readability. Thus, PCR and LAMP assays are called “markers”, alleles of the DEP1 gene are called sometimes “genes”, etc.

2.      Authors are encouraged add in the Introduction section information about exact molecular markers that they intended to detect using LAMP. More details about other dep1 variants would be also welcomed.

3.      Sequence alignments used for primer selection need to be presented in the manuscript for readers to see how are these areas actually conservative.

4.      More details of PCR and LAMP are necessary, such as primer concentrations, template amount or concentration, exact temperature profiles, etc. Without them, repeating of the presented results could be impossible.

5.      Additional loop primers are known to increase LAMP speed and sensitivity. Have authors considered design of loop primers in the reported assay?

6.      LAMP is highly prone to contamination due to high yield of products. Authors used gel-electrophoresis for optimization of LAMP conditions which could lead to contamination of labware and reagents. Countermeasures prevented contamination are important in these regards and need to be specified. Also, authors are encouraged to provide results for no-template controls in each figure.

7.      The developed assay doesn’t provide a possibility to ensure presence of rice genomic DNA in analyzed samples. Thus, it is possible to miss samples with the desirable allele because of issues with DNA purification.

8.      Authors are encouraged to specify analytical sensitivity of the assay using control samples with known DNA concentrations and compare it with other known techniques.

9.      The Discussion section seems to miss information about other reported tests for the dep1 allele which is necessary for correct assessment of the designed LAMP test. Also, limitations of LAMP, e.g., contamination issues, false-positive results, etc, need to be mentioned.

Minor issues:

1.      Page 2, line 66 — LAMP was reported in 2000.

2.      Figures 1 and 2 are absent in the manuscript.

3.      Legends of figures 7 and 8 seem to be mixed up in places.

Comments on the Quality of English Language

About language, please, refer to the first comment in the Comments section.

Author Response

(The authors gave the same response as above.)

Reviewer 3 Report

Comments and Suggestions for Authors

Major Revisions:

- The introduction provides extensive background on dep1, but lacks clarity on the specific gap or problem being addressed by developing a visual LAMP detection method. Please clearly state the limitations of existing methods and how your approach aims to address that gap.

- The results heavily focus on validating the accuracy of the detection method, but more data demonstrating its application for breeding or field use would strengthen the paper. Consider expanding the section on incorporating dep1 into SHUHUI498.

- The discussion could be expanded to provide more context, implications and future directions for using rapid visual detection of dep1. Elaborate on how this method can aid breeding programs and development of improved varieties.

Minor Revisions:

- Some acronyms like LAMP and CAPS are used without being defined on first use. Please spell out abbreviations.  

- Carefully proofread the manuscript to fix minor grammar issues, awkward phrasings, and formatting inconsistencies.

- The quality of figures 2 and 5 could be improved with higher resolution and clarity.

- In methods, elaborate a bit more on the software used for analysis and primer design.  

- Consider condensing parts of the long introduction that describe dep1 effects by citing more reviews.

- Discussion could comment on any limitations or potential improvements to the detection method.

Author Response

Comments 1: Major Revisions:

- The introduction provides extensive background on dep1, but lacks clarity on the specific gap or problem being addressed by developing a visual LAMP detection method. Please clearly state the limitations of existing methods and how your approach aims to address that gap.

Response 1: Thank you for your kind comment. We have included a thorough introduction on the detection methods use with their limitation for practical use and the advantages offered by LAMP method in comparison (“Several reports have been presented on the detection of dep1 allele using PCR (Polymerase Chain Reaction) technology based on molecular markers such as InDel-E5 [17], DEP1S9 [18], DEP1E5ID [19] and H90 [20]. However, these methods require high temperature and specialized devices for data visualization, rendering it impractical for on-site testing [21,22]. LAMP (loop-mediated isothermal amplification) is a isothermal nucleic acid detection technique [23,24] that is able to achieve an accuracy similar to PCR. Moreover, the technology does not require specialized nor sophisticated instru-ment while offering rapid amplification of targeted DNA with high efficiency, specific-ity and sensitivity. LAMP technology is also user-friendly as it can be easily mastered and performed by laymen without the need for prior molecular experiences [25]. Re-cently, LAMP has been successfully used for rapid detection of transgenes [26], blight resistant genes [27], and authenticity [28] et al. in rice plants.” Line 59-70).

Comments 2: - The results heavily focus on validating the accuracy of the detection method, but more data demonstrating its application for breeding or field use would strengthen the paper. Consider expanding the section on incorporating dep1 into SHUHUI498.

Response 2: Thank you for your kind comment. The incorporation of dep1 into SHUHUI498 has been already been described in the manuscript (please refer to sections 2.4, 3.4, and 3.5)

Comments 3: - The discussion could be expanded to provide more context, implications and future directions for using rapid visual detection of dep1. Elaborate on how this method can aid breeding programs and development of improved varieties.

Response 3: Thank you for your kind suggestion. We have revised the manuscript to further illustrate how the development of a colorimetric LAMP assay would aid the marker-assisted breeding programs (“The colorimetric LAMP assay for dep1 allele designed in this study, combined with the rapid DNA extraction methods [26,46], would present a convenient method for on-site genotyping of dep1, contributing greatly to the breeding and development of green, high-yielding rice varieties.”, line 359-362), and addition of future development of the assay in the conclusion (“For further study, we would investigate the incorporation of additional loop primers to our LAMP assay in order to boost their efficiency.”, line 392-393), among others.

Comments 4: Minor Revisions:

- Some acronyms like LAMP and CAPS are used without being defined on first use. Please spell out abbreviations.  

Response 4: Noted with thanks. We have made the changes accordingly.

Comments 5: - Carefully proofread the manuscript to fix minor grammar issues, awkward phrasings, and formatting inconsistencies.

Response 5: Noted with thanks. We have revised the manuscript to improve the English and eliminate inconsistencies.

Comments 6: - The quality of figures 2 and 5 could be improved with higher resolution and clarity.

Response 6: Thank you for your kind suggestion. Unfortunately, that is the highest resolution we have.

Comments 7: - In methods, elaborate a bit more on the software used for analysis and primer design.  

Response 7: Thank you for your kind suggestion. We have included the link for the software used for analysis and primer design (line 90-96).

Comments 8: - Consider condensing parts of the long introduction that describe dep1 effects by citing more reviews.

Response 8: Thank you for your kind suggestion. However, we feel that explaning the functions and beneficial effects of dep1 are vital to justify our study.

Comments 9: - Discussion could comment on any limitations or potential improvements to the detection method.

Response 9: Thank you for your kind suggestion. We have revised the discussion to better reflect our argument for LAMP assay, including mentioning the main limitation (“Even though LAMP protocols are prone to false positive due to their high sensitivity, …” line 338-339), comparison with conventional PCR ( “The results from our study have indicated that our developed LAMP assay has comparable precision and sensitivity to standard PCR using InDel markers (Figure 3 and Figure 8A), further lending credence to the potential of colorimetric LAMP assay.”, line 348-351), and future development (“For further study, we would investigate the incorporation of additional loop primers to our LAMP assay in order to boost their efficiency.”, line 388-389).

Round 2

Reviewer 1 Report

Comments and Suggestions for Authors

The authors have attended the comments of the reviewers 

Comments on the Quality of English Language

Good

Author Response

Comments 1: The authors have attended the comments of the reviewers。

Response 1: Thank you for your kind and valuable comments

Reviewer 2 Report

Comments and Suggestions for Authors

The revised manuscript was thoroughly corrected and most arisen questions are replied. Author’s efforts are greatly appreciated. However, several issues still need to be clarified.

Major issues:

1.      Many thanks to authors for language editing. However, some points seem to be confusing. Plausibly, the term “marker” in the case of amplification assays means a structural polymorphism of DNA or RNA detected by the assay.

2.      Indeed, amplification products are presented in figures with PCR results. However, a possibility of inhibition or absence of DNA in analyzed samples needs to be considered. In other case, false-negative results can be obtained, if no LAMP product accumulates after the reaction due to inhibition or low DNA template concentration. To ensure DNA quality and quantity, amplification tests commonly target a locus on genomic DNA that is detected despite the sample’s genotype.

3.      Authors are encouraged to add information about necessary positive control samples for the designed assay. Such samples are essential part of any test as they allow to control reagent’s quality. Real LAMP sensitivity can vary in a broad range depending on target and primer’s set. In that sense, if empirical determination of analytical sensitivity is not possible, the inner control and standard control sample are missing, it needs to be mentioned as limitations of the study.

Minor issues:

1.      Page 3, line 119: “4 µL of 2 × Taq PCR mix” — was 4 µL of a 2 × PCR mix used for 20 μL reaction?

2.      Page 3, lines 133-134: “10 µM F3/B3 primers (each 1 µL), 10 µM FIP/BIP primers (each 0.25 µL)” — primer concentrations seem to be lower than recommended for LAMP. Also, concentration of F3/B3 primers seem to be higher than that of FIP/BIP primers. 

Author Response

Comments 1: The revised manuscript was thoroughly corrected and most arisen questions are replied. Author’s efforts are greatly appreciated. However, several issues still need to be clarified.

Major issues:

  1. Many thanks to authors for language editing. However, some points seem to be confusing. Plausibly, the term “marker” in the case of amplification assays means a structural polymorphism of DNA or RNA detected by the assay.

Response 1: Thank you for pointing this out. We agree with this comment. Therefore, we have changed the term “marker” in the captions to DNA ladder (in the captions, line 213, 225, 236, 266, 305) or primers (in text, line 20, 22, 175) where applicable.

Comments 2:  Indeed, amplification products are presented in figures with PCR results. However, a possibility of inhibition or absence of DNA in analyzed samples needs to be considered. In other case, false-negative results can be obtained, if no LAMP product accumulates after the reaction due to inhibition or low DNA template concentration. To ensure DNA quality and quantity, amplification tests commonly target a locus on genomic DNA that is detected despite the sample’s genotype.

Response 2: Thank you for your kind comment. We would like to provide a point-by-point reply to the concerns raised:

  1. The DNA templates were checked prior to PCR using Nanodrop to ensure necessary purity and quantity. With confirmed DNA quality, PCR can be run without internal control and negative controls (references: Figure 3 from DOI: 10.1038/s41598-023-29730-6; Figure 4 from DOI: 10.1007/s11032-015-0191-y);
  2. While inhibition of PCR is possible, this is often caused by the presence of PCR inhibitors, and we did not see any results that suggest that this had happened in our experiments, considering the clear and bright bands we obtained in Figure 3, which also eliminate the possibility of DNA absence in our extracted samples. Furthermore, available literature (doi:10.3390/cells10081931) shows that the LAMP reaction is more tolerant to inhibition than conventional PCR;
  3. In regards to the concern of false-negative results, while we agree with the reviewer of the need to ensure its prevention, point 1 and 2 explained that we are unlikely to have false-negative results, which is illustrated in Figure 5A, which shows every sample producing expected band that is aligned to their known genotype. However, we understanding the perspective of the reviewer and would address the issue in text (see response 3).

Comments 3:  Authors are encouraged to add information about necessary positive control samples for the designed assay. Such samples are essential part of any test as they allow to control reagent’s quality. Real LAMP sensitivity can vary in a broad range depending on target and primer’s set. In that sense, if empirical determination of analytical sensitivity is not possible, the inner control and standard control sample are missing, it needs to be mentioned as limitations of the study.

Response 3: Thank you for your kind suggestion. We agree that positive controls play an important role in checking the quality of reagents. Hence, we would include the design of an internal control and standard control in further development of our assay into commercial kit(s). Therefore, we have added the following statement in the discussion: “It is important to note that while the results in our study are in agreement with each other and are fully aligned with our expected outcome, the experiments were nevertheless conducted in a controlled environment with known genotypes. For the assay to be applicable for field use which involves many unknown variables, internal and standard controls have to be developed to ensure the accuracy, including prevention of false-negative results. (line 383-388)”

Comments 4: Minor issues:

  1. Page 3, line 119: “4 µL of 2 × Taq PCR mix” — was 4 µL of a 2 × PCR mix used for 20 μL reaction?
  2. Page 3, lines 133-134: “10 µM F3/B3 primers (each 1 µL), 10 µM FIP/BIP primers (each 0.25 µL)” — primer concentrations seem to be lower than recommended for LAMP. Also, concentration of F3/B3 primers seem to be higher than that of FIP/BIP primers

Response 4: Noted with thanks. “a” was added. The concentration difference is deliberate, as instructed by the commercial kit we used.